# [Re] GNNBoundary: Towards explaining Graph Neural Networks through the lens of decision boundaries

## Abstract

Graph Neural Networks (GNNs) have been successfully applied to machine-learning tasks for graph-structured data. However, their decision-making process remains difficult to interpret. The GNNBoundary method proposed by Wang & Shen (2024) is a model-level explanation method designed to analyze GNN decision boundaries. This study aims to reproduce and verify the claims made in the original paper: (1) GNNBoundary method can identify the adjacent classes, (2) GNNBoundary method can generate faithful near-boundary graphs, and (3) these graphs can be used to analyze the decision boundary. Experiments were conducted on four datasets, including the Proteins dataset, which extends the original work. To reproduce the results, we followed the authors' open-sourced implementation. Our findings only partially support Claim 1, due to variations found in adjacent classes. Generally, we were able to generate faithful near-boundary graphs, mostly supporting Claim 2. The boundary analysis differed from the original results, but it was in line with results for adjacent classes and confusion matrices, partially verifying Claim 3. Further support for this claim was found on Proteins through a PCA visualization of the data.

## 1 Introduction

Graph Neural Networks — GNNs (Scarselli et al., 2009) are a class of deep learning methods used for modelling graph-structured data such as molecules, social networks, and recommendation systems. Three primary tasks performed by GNNs are node classification (Maekawa et al., 2022), graph classification (Errica et al., 2022) and link prediction (Zhang & Chen, 2018). Although GNNs are widely applied, their decision-making process remains rather poorly understood, as the decision boundaries can be very complex. Consequently, trustworthiness of GNN models is difficult to assess and poses a significant challenge, especially while making decisions in high-stakes domains such as healthcare (Ahmedt-Aristizabal et al., 2021).

To address the need for transparency and interpretability of GNNs, novel methods have been introduced. They can be divided into two groups, namely instance-level explanations (Ying et al., 2019; Luo et al., 2020) and model-level explanations (Yuan et al., 2020; Wang & Shen, 2023). The former are focused on explaining the outcome of the GNN for specific input. It means that these methods address individual examples rather than the overall behaviour of the model.

On the other hand, model-level explanations aim to explore the general behaviour of the model, such as its decision boundaries, without any particular input. However, there are three main limitations with current model-level explainability methods. First, they do not investigate which classes are more likely to get confused with one another. Second, the circumstances that make the GNN less certain about its predictions are not well investigated. Finally, it is not trivial to obtain a decision-making scheme that captures the complex inter-class relationships.

One of the recent works about model-level explainability methods is 'GNNBoundary: Towards explaining Graph Neural Networks through the lens of decision boundaries' (Wang & Shen, 2024), which aims to explain the decision boundary for graph classification tasks. The first step in GNNBoundary is to identify adjacent classes, as decision boundaries can only exist between these classes. Next, for each adjacent class pair, a near-boundary graph is generated. The original study evaluated the GNNBoundary method quantitatively

on different datasets and showed that it can convey key boundary characteristics. In this paper, these findings are analyzed and further refined with similar experiments on an additional dataset.

## 2 Scope of reproducibility

The aim of this paper is to reproduce the results of Wang & Shen (2024) and verify the following claims presented in the GNNBoundary paper:

- **Claim 1**: The GNNBoundary method effectively identifies adjacent classes, where decision boundaries are most likely to exist.

- **Claim 2**: The GNNBoundary method generates faithful boundary graphs that accurately represent decision boundaries.

- **Claim 3**: Generated boundary graphs can be used to analyze decision boundaries by measuring: boundary margin, boundary thickness, and boundary complexity. These metrics provide insights into the decision-making process of GNNs.

## 3 Background

In general, a graph is written as $G = (\mathcal{V}, \mathcal{E})$, where $\mathcal{V}$ is the node set and $\mathcal{E}$ is the edge set. Adjacency matrix $\boldsymbol{A} \in \{0, 1\}^{N \times N}$, where $N$ is the number of nodes, is used to store information about node adjacency. Node feature matrix $\boldsymbol{Z} \in \mathbb{R}^{N \times d}$, where $d$ is the number of features for each node, encodes information about node attributes. For GNN with $L$ layers, the embedding function composed of first $l$ layers with input $G$ is denoted as $\phi_l$, and $\eta_l$ indicates the scoring function corresponding to $L - l$ layers before softmax.

GNN consists of three main operations: computing messages, aggregating messages and updating the hidden node representations $\mathbf{H}^{(l)} = \phi_l(G)$ (Wang et al., 2021).

When the graph passes through the classifier, the input space as well as the embedding space is divided into $C$ decision regions $\{\mathcal{R}_c^{(l)} \mid c \in [1, C]\}$ for each layer $l$. The role of the classifier is to predict the class $c$ of graph $G$, where $c = \operatorname{argmax}_k f_k(G)$ where $k$ denotes the index of the class among $C$ possible classes. The decision boundary is a region in a space where the classifier is uncertain between two classes $c_1$ and $c_2$ i.e. $G$ has an equal probability of belonging to both classes. The decision boundary is defined as $\mathcal{B}_{c_1 \| c_2}^{(l)} = \{\mathbf{H}^{(l)} : \sigma(\eta(\mathbf{H}^{(l)}))_{c_1} = \sigma(\eta(\mathbf{H}^{(l)}))_{c_2} > \sigma(\eta(\mathbf{H}^{(l)}))_{c'}, \forall c' \neq c_1, c_2\}$.

## 4 GNNBoundary

GNNBoundary is a novel model-level explainability method designed to analyze the decision boundaries of GNNs. The method has three main stages: (i) identifying adjacent classes, (ii) generating near-boundary graphs, and (iii) analyzing decision boundaries using various metrics.

### 4.1 Identifying adjacent classes

Identifying adjacent classes is a crucial step to establish the location of the decision boundaries and to find a boundary graph $G_{c_1 \| c_2} \in \mathcal{B}_{c_1 \| c_2}$. The boundary graph can exist only if the boundary embeddings $\mathbf{H}_{c_1 \| c_2}^{(l)} \in \mathcal{B}_{c_1 \| c_2}^{(l)}$ are present. Therefore, information about boundary graphs can be inferred from the embedding space, as it better reflects the separation between classes. In order to determine the likelihood of the boundary graph, we calculate the ubiquity of embeddings. Ubiquity can be determined as a probability of $\mathbf{H}_{c_1 \| c_2}^{(l)}$ appearing between the embedding regions $\mathcal{R}_{c_1}^{(l)}$ and $\mathcal{R}_{c_2}^{(l)}$. To simplify calculations, only embeddings of the last hidden layer are taken into consideration and in this case ubiquity of $\mathbf{H}_{c_1 \| c_2}^{(L-1)}$ is referred to as degree of adjacency. Two classes are adjacent if the degree of adjacency exceeds a predefined threshold.

In the original paper, Algorithm 1 (Appendix A) is proposed to measure the degree of adjacency. First, we randomly sample K pairs of graphs $G_{c_1}$ and $G_{c_2}$ using Monte Carlo sampling. To calculate the degree of adjacency, linear interpolation is done between $\mathbf{H}_{c_1}^{(L-1)}$ and $\mathbf{H}_{c_2}^{(L-1)}$. Given that the decision boundary is linear, it means determining if there is any other decision region between $\mathbf{H}_{c_1}^{(L-1)}$ and $\mathbf{H}_{c_2}^{(L-1)}$. The degree of adjacency is then computed as a ratio of adjacent embeddings found in $K$ samples. As a result of this process, an adjacency degree matrix is generated, which identifies the possible adjacent classes.

## 4.2 Learning objective

Once adjacent classes are identified, boundary graphs can be generated. In order to do so effectively, there are two desired properties that need to be satisfied by the objective function:

- Property 1: For boundary classes ($b \in \{c_1, c_2\}$), objective should encourage posterior probability $p(b)$ if $p(b) < 0.5$ and discourage $p(b)$ if $p(b) \geq 0.5$ to ensure balance between the adjacent classes.

- Property 2: Objective function should also encourage high logit values $f(G)_b$ for boundary classes and discourage high logit values $f(G)_{b'}$ for other classes $b'$. This ensures that the boundary graph is focused on two boundary classes.

In the original paper, a novel objective function $\mathcal{L}$ is proposed that satisfies both properties:

$$\min_G \mathcal{L}(G) = \min_G \sum_{b' \notin \{c_1, c_2\}} \beta f(G)_{b'} \cdot p^*(b')^2 - \sum_{b \in \{c_1, c_2\}} \alpha f(G)_b \cdot \left(1 - p^*(b)\right)^2 \cdot \mathbf{1}_{p^*(b) < \max_{c \in [1,C]} p^*(c)}$$

where the first term penalizes high logits and probabilities for non-boundary classes, while the second term ensures that boundary classes are well-balanced. Constants $\alpha$ and $\beta$ are hyperparameters, and $p^*(b)$ and $p^*(b')$ are target class probabilities, where the former should be around 0.5 and the latter close to 0.

It is assumed that the boundary graphs are Gilbert random graphs (Gilbert, 1959) and that node features are independently distributed. To generate a boundary graph, we sample from the following boundary graph distribution:

$$P(G) = \prod_{v_i \in \mathcal{V}} P(z_i) \cdot \prod_{(v_i, v_j) \in \mathcal{E}} P(a_{ij})$$

where $z_i$ stands for node feature of node $v_i$ and $a_{ij} = 1$ if nodes $v_i$ and $v_j$ are connected and it is zero otherwise. $P(z_i)$ is a probability distribution over node features and $P(a_{ij})$ stands for probability distribution of connections between nodes. Usually $a_{ij}$ and $z_i$ are taken such that $a_{ij} \sim \text{Bernoulli}(\theta_{ij})$ and $z_i \sim \text{Categorical}(\mathbf{p}_i)$ with $\|\mathbf{p}_i\|_1 = 1$. However, to optimize the objective function we need $\nabla_A \mathcal{L}(G)$, which doesn't exist as the graph data is discrete. To allow standard optimization techniques, a reparametrization trick with continuous relaxation of graphs is introduced. Discrete variables are now modelled as continuous in the following way:

$$\begin{cases} \tilde{\mathbf{z}}_i \sim \text{Concrete}(\zeta_i, \tau_z) & \text{for } \tilde{\mathbf{z}}_i \in [0,1]^d, \|\tilde{\mathbf{z}}_i\|_1 = 1 \text{ and } \zeta_i \in \mathbf{Z}, \\ \tilde{a}_{ij} \sim \text{BinaryConcrete}(\omega_{ij}, \tau_a) & \text{for } \tilde{a}_{ij} \in [0,1] \text{ and } \omega_{ij} \in \mathbf{\Omega}. \end{cases}$$

To allow for differentiable sampling procedure, Gumbel-Softmax trick (Jang et al., 2017) is implemented in the following way:

$$\begin{cases} \tilde{\mathbf{z}}_i = \text{Softmax}\left((\zeta_i - \log(-\log \epsilon))/\tau_z\right), \\ \tilde{a}_{ij} = \text{sigmoid}\left((\omega_{ij} + \log \epsilon - \log(1 - \epsilon))/\tau_a\right) \end{cases}$$

where $\tau_z$ and $\tau_a$ are hyperparameters that control approximation of Categorical distribution and $\epsilon \sim \text{Uniform}(0, 1)$.

Therefore, the distribution of boundary graphs is learned using the following objective:

$$\min_{\mathbf{A},\mathbf{Z}} \mathcal{L}(G) = \min_{\mathbf{\Theta},\mathbf{P}} \mathbb{E}_{G \sim P(G)}[\mathcal{L}(\mathbf{A}, \mathbf{Z})] \approx \min_{\mathbf{\Omega},\mathbf{Z}} \mathbb{E}_{\epsilon \sim U(0,1)}[\mathcal{L}(\tilde{\mathbf{A}}, \tilde{\mathbf{Z}})] \approx \min_{\mathbf{\Omega},\mathbf{Z}} \frac{1}{K} \sum_{k=1}^{K} \mathcal{L}(\tilde{\mathbf{A}}, \tilde{\mathbf{Z}})$$

where $\tilde{\mathbf{A}}$ and $\tilde{\mathbf{Z}}$ are continuously relaxed versions of adjacency matrix $\boldsymbol{A}$ and node feature matrix $\boldsymbol{Z}$.

### 4.3 Training GNNBoundary

In theory, the training procedure (Algorithm 2 (Appendix A)) should stop when a boundary graph is generated that belongs to the decision boundary $\mathcal{B}_{c_1\|c_2}$, i.e. $\sigma(f(G))_{c_1} = \sigma(f(G))_{c_2} = 0.5$. However, such a strict stopping criterion is not feasible. Instead, a more relaxed, near-boundary criterion[1] is defined as follows:

$$\Psi(G) = \mathbf{1}_{p(c_1),p(c_2)\in[p_{\min},p_{\max}]}(G).$$

Where $p_{\min}$ and $p_{\max}$ are the lower and upper bounds, respectively, within which the class probabilities $p(c_1)$ and $p(c_2)$ must lie for a graph to be considered near the decision boundary. Standard regularization techniques, $L_1$ and $L_2$ regularization, are implemented to prevent overfitting as well as gradient saturation problems. Moreover, to impose desired design of boundary graphs budget penalty $R_{\text{budget}}$ is implemented, which encourages succinct graphs i.e. with manageable number of edges:

$$R_{\text{budget}} = \text{Softplus}\left(\|\text{sigmoid}(\mathbf{\Omega})\|_1 - B\right)^2.$$

Finally, a dynamic regularization scheduler is implemented to address the problems with convergence that might be caused by budget penalty. The scheduler adjusts the budget penalty weight during training. In the early stages, the budget penalty weight is smaller and it increases as the graph approaches the boundary. This ensures that the budget penalty is balanced throughout the training procedure. For iteration $t$ budget penalty weight is defined as:

$$w_{\text{budget}}^{(t)} = w_{\text{budget}}^{(t-1)} \cdot s_{\text{inc}}^{\mathbf{1}\{\Psi(G^{(t)})\}} \cdot s_{\text{dec}}^{\mathbf{1}\{\neg\Psi(G^{(t)})\wedge(s_{\text{dec}}\cdot w_{\text{budget}}^{(t-1)}\geq w_{\text{budget}}^{(0)})\}},$$

where $w_{\text{budget}}^{(0)}$, $s_{\text{inc}}$ and $s_{\text{dec}}$ are hyperparameters for initial budget weight, weight increment and weight decrement and $G^{(t)} = \mathbb{E}_{G\sim P(G)}[G]$ for iteration $t$.

### 4.4 Boundary analysis

To gain more insight into the decision-making process of GNNs three metrics are used: boundary margin, boundary thickness and boundary complexity. Besides boundary graphs, graphs for each class, $G_c \in \mathcal{R}_c$, are also needed for analysis. In the original paper, authors used graphs generated by GNNInterpreter rather than those from training data as it makes no assumption about accessibility of training data and by doing so analysis can extend beyond the in-distribution data.

Boundary margin measures how far the graph for class $c$ is from the boundary graph (i.e. decision boundary) and it is calculated in the following way:

$$\Phi(f,c_1,c_2) = \min_{(G_{c_1},G_{c_1\|c_2})}\left\|\phi_l(G_{c_1}) - \phi_l(G_{c_1\|c_2})\right\|$$

where $\phi_l$ is the graph embedding from the graph pooling layer. Larger boundary margin means that the classifier has better performance, as it can make a clear distinction between classes. Moreover, larger margin also suggests better robustness to graph perturbations.

The formula for boundary thickness is:

$$\Theta(f,\gamma,c_1,c_2) = \mathbb{E}_{(G_{c_1},G_{c_1\|c_2})\sim P}\left[\left\|\phi_l(G_{c_1}) - \phi_l(G_{c_1\|c_2})\right\|\int_0^1 \mathbf{1}_{\gamma>\sigma(\eta_l(h(t)))_{c_1}-\sigma(\eta_l(h(t)))_{c_2}}\,dt\right]$$

where $h(t) = (1-t)\cdot\phi_l(G_{c_1}) + t\cdot\phi_l(G_{c_1\|c_2})$ for $t\in[0,1]$ and $\gamma$ is a hyperparameter. Thicker decision boundaries result in a more robust model, as small perturbations to the input data are less likely to influence the output of the classifier.

---

[1]From here on we use 'near-boundary graph' and 'boundary graph' interchangeably.

The last measure to analyze a decision boundary is boundary complexity defined as:

$$\Gamma(f, c_1, c_2) = \frac{H(\lambda/\|\lambda\|_1)}{\log D} = \frac{-\sum_i (\lambda_i/\|\lambda\|_1) \log (\lambda_i/\|\lambda\|_1)}{\log D}$$

where $\lambda$ are the eigenvalues of the covariance matrix of $\mathbf{X}_{c_1\|c_2}$, and $\mathbf{X}_{c_1\|c_2} \in \mathbb{R}^{|B_{c_1\|c_2}| \times D}$ is formed by the $\phi_{L-1}(G_{c_1\|c_2})$, $\forall G_{c_1\|c_2} \in B_{c_1\|c_2}$. A complex decision boundary may indicate that it is overfitting to the training data, which can lower generalizability to unseen data. A simpler decision boundary is expected to capture the most important patterns in data, disregarding irrelevant noise in training data, and hence generalize better.

## 5 Methodology

### 5.1 Datasets

In the original paper, three datasets (link to download) are used to conduct experiments: one synthetic dataset and two real-world datasets. For the synthetic dataset Motif (Wang & Shen, 2023) graphs are labelled based on the motifs they contain - House, House-X, Comp_4 and Comp_5. A separate class is designed for graphs without any motif. Each node in the graphs has a feature with one of five possible values (colours). The training set of Motif contains 10378 graphs, while the test set comprises 1153 graphs.

The first real-world dataset used is Collab (Yanardag & Vishwanathan, 2015), which represents the relationships between researchers in multiple different fields. Classes in Collab are related to three areas in physics: High Energy (HE), Condensed Matter (CM) and Astro. Each graph in Collab is an ego network which means that it is centered around a specific researcher and shows their interactions with other researchers across fields. The training set comprises 4500 graphs and besides that 500 graphs are used for testing.

The second real-world dataset used is Enzymes (Borgwardt et al., 2005), which consists of 600 enzymes taken from the BRENDA enzyme database (Schomburg et al., 2004), which are classified into six classes. Each enzyme is represented as a graph with three possible types of nodes. The dataset is split into a training set with 540 graphs and a test set with 60 graphs.

We extend the original study by introducing the Proteins dataset (Borgwardt et al., 2005), which contains graphs representing amino acid interactions, with nodes indicating amino acids and edges reflecting their interactions. This dataset, comprising two classes—Enzyme and Non-Enzyme—was chosen due to the challenges faced when reproducing the results of the Enzymes dataset, which shares structural similarities. Unlike Enzymes, the Proteins dataset is a more established benchmark in the graph neural network community, with well-documented properties. By incorporating this dataset, we seek to explore whether the difficulties with Enzymes arise from the model's sensitivity to particular biological structures or dataset-specific nuances, ultimately contributing to a more comprehensive evaluation of GNNBoundary's performance.

#### 5.1.1 GCN architectures

In the original study, a Graph Convolutional Network (GCN) (Kipf & Welling, 2017) is employed to distinguish between graph classes across the Motif, Collab, and Enzymes datasets. The Motif classifier is composed of 3 layers with 6 hidden channels, while the Collab classifier uses 5 layers with 64 hidden channels. For the Enzymes and Proteins datasets, the GCN is configured with 3 layers and 32 hidden channels. Each model applies LeakyReLU activations between layers, followed by sum and mean pooling of the embeddings. The pooled embeddings are then concatenated and passed through a linear layer with a ReLU activation to produce the final embeddings, which serve as the input for the classification task.

### 5.2 Hyperparameters

To reproduce the results presented in the original GNNBoundary paper, we initially applied the hyperparameters from the Jupyter notebooks provided in the repository. However, discrepancies emerged between the reported results and our reproduced findings. To address this, we reverted to the exact hyperparameter

values specified by the authors wherever possible. This involved using $\tau = 0.15$ as the temperature parameter for the Concrete Distribution, a sample size of K = 32 for each Monte Carlo sampling, and an initial learning rate of 1 for the SGD optimizer. The learning rate was dynamically adjusted throughout training via the scheduler provided in the repository.

Furthermore, for each adjacency pair in the Collab, Motif, and Enzymes datasets, we tuned hyperparameters accordingly to identify the optimal configuration required to reproduce the paper's results. Due to time constraints for this project, the tuning process primarily focused on the regularization weights $R_{L1}$, $R_{L2}$, and the budget value $B$. The hyperparameters $R_{L1}$ and $R_{L2}$ were selected for their role in mitigating the saturating gradient problem, while $B$ was chosen for its direct influence on the budget penalty $R_{budget}$, which encourages the generation of succinct boundary graphs (Wang & Shen, 2024). Additionally, since the performance of GNNInterpreter is influenced by random initialization (Vasilcoiu et al., 2024) and GNNBoundary's code is built upon GNNInterpreter, we fixed the seed to 3407 for all experiments (Picard, 2021). Other hyperparameters were manually tuned to maximize the convergence speed and success rate. The tuned hyperparameter values can be found in (Appendix B, Table 2).

### 5.3 Experimental setup and code

The code used to replicate and extend the experiments can be found in our GitHub repository (https://anonymous.4open.science/r/FACT-AI-2025-A647/). We include instructions on how to run all experiments for each dataset. The code provided by Wang & Shen (2024) included the main logic and functionality required to conduct their experiments, but not the overall setup procedure described in the paper. Given this, we found it necessary to implement additional features and adjust the code structure.

**GCN classifier.** We validate the performance of the GCN model by testing both the provided model checkpoints and training from scratch. We begin by testing the existing checkpoints and running the analysis with the hyperparameter values from the Jupyter notebook demos, followed by those from the paper. Ultimately, we retrain the GCN classifier from scratch with the specified hyperparameters for consistency and conduct the remainder of our study.

**Class pair adjacency.** To verify Claim 1, we adopt the implementation of (Wang & Shen, 2024), with a modification to use graphs sampled directly from the datasets instead of GNNInterpreter. In the original paper, this is noted as a viable alternative for conducting boundary analysis. Given the unreliable performance of GNNInterpreter in generating graphs (Vasilcoiu et al., 2024), and the lack of access to the relevant code, this approach proved to be both more efficient and consistent with the replicability of the study. The core analysis remains unchanged: the likelihood of adjacency between two classes is quantified using a ubiquity measure. This measure calculates how often the embeddings of one class transition into the decision region of another class along a straight path. By randomly sampling 100 graph pairs from the two classes and interpolating their embeddings, the degree of adjacency is determined as the proportion of sampled pairs that satisfy the boundary condition. If this proportion is 0.8 or above, a class pair is considered adjacent.

**GNNBoundary.** Validating the claim that GNNBoundary generates faithful boundary graphs (Claim 2) required provisions beyond the training procedure outlined by the authors. Neither the parameters provided in the paper nor those in the code led to convergence of the training procedure for class pairs, let alone generation of faithful boundary graphs. This prompted hyperparameter tuning in the form of a grid search over the regularization weights as outlined above. Hyperparameters were deemed optimal if they resulted in convergence during training. With this in mind, we track the success rate within 500 iterations and average convergence iteration over 1000 runs as in the original study. Upon successful training, we sample 500 boundary graphs and report their mean class probabilities for the adjacent classes along with the standard deviation. Generated graphs are further used to investigate Claim 3.

**Boundary Analysis.** With the sampled boundary graphs, we calculate the outlined boundary metrics to gain deeper insight into the decision boundaries of adjacent classes (Claim 3). The author's original repository did not contain implementations for these, such that boundary margin (Yang et al., 2020), thickness (Yang et al., 2020), and complexity (Guan & Loew, 2020) were implemented by ourselves. With these metrics

computed and our knowledge of the datasets for which boundary graphs were generated, we proceed to discuss the properties of the decision boundaries.

## 5.4 Computational requirements

The experiments were run on a single CPU, a AMD Ryzen 9 7900X. The environment was replicated as closely as possible to the one specified in the authors' repository of the original paper. We did not adhere to the versions specified in the original authors' paper, as the versions of PyTorch and PyG mentioned did not satisfy certain package requirements. To accelerate the training of the GCN classifier from scratch, a GPU was employed. The CUDA version utilized was 12.1, running on an NVIDIA GeForce RTX 4090 GPU. To conduct the experiment in Table 5 (Appendix D) for a given pair of classes, the minimum energy consumption required is 0.020 kWh over a duration of 9 minutes and 15 seconds, while the maximum energy consumption required is 0.264 kWh over a duration of 2 hours, 2 minutes, and 44.8 seconds. On average, testing each pair in Table 1 requires $4.795e-5$ kWh over a duration of 1.6 seconds.

# 6 Results

## 6.1 Results reproducing original paper

**Adjacency Analysis** To verify the precision of the adjacency analysis proposed in the original GNNBoundary method, we sought to replicate the adjacency findings (Appendix C, Figure 6).

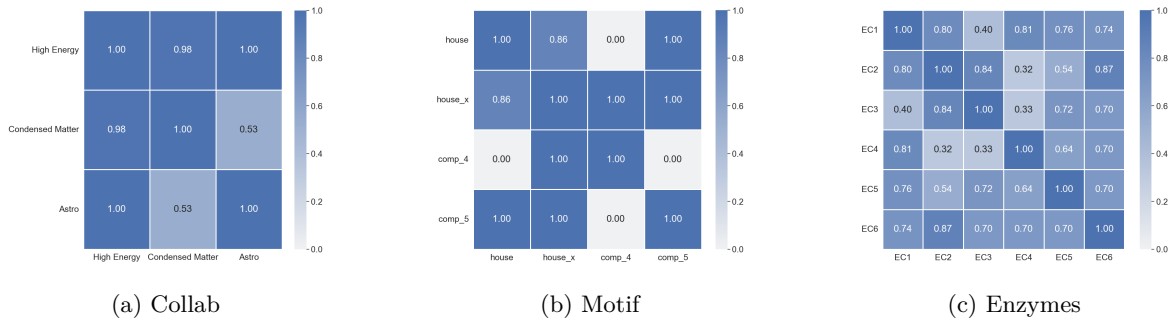

(a) Collab        (b) Motif        (c) Enzymes

Figure 1: The degree of adjacency of every class pair. High values ($\geq 0.8$) indicate the decision regions of the classifier are adjacent.

Using an adjacency threshold of 0.8, our results (Figure 1) aligned with the original adjacent class pairs for the Collab dataset but diverged for Motif and Enzymes (Appendix C, Figure 6). Specifically, we did not identify (House, Comp_4) in Motif as adjacent but found (House, Comp_5) and (House_X, Comp_4) to be adjacent, unlike the original study. For Enzymes, EC1-EC5, EC1-EC6, EC4-EC5, and EC5-EC6 were not recognized as adjacent, while EC1-EC2 and EC2-EC6 were identified as adjacent. Notably, the adjacency scores for EC1-EC5, EC1-EC6, and EC2-EC6 were close to the 0.8 threshold, highlighting the sensitivity of classification. Therefore, Claim 1 is partially supported.

**Boundary Graph Generation.** The results in Table 1 concur with those of the original study in that the probabilities of the sampled graphs for adjacent class pairs outperform the random baseline. The probabilities themselves generally differ from those reported in the original study often falling out of the 0.45-0.55 ($p_{min} - p_{max}$) range, with a maximum discrepancy of 23.1% (Appendix C, Table 4), despite the fact that the near-boundary criterion being satisfied during training. For some pairs, the near-boundary criterion was relaxed from the probability range of 0.45-0.55 to 0.4-0.6 (Appendix B) to facilitate convergence for the GNNBoundary sampler. In the case of the House-Comp_4 pair, no convergence was achieved during reproduction (Appendix D, Table 5). The Collab dataset exhibits the best performance overall, with small standard errors across both adjacent pairs and probabilities within the desired 0.45-0.55 range. Satisfactory results are only achieved for one pair in Motif (HouseX and Comp_5) and Enzymes (EC5-EC6) respectively,

although some probabilities fall slightly out of range. Ultimately, we can only say that faithful boundary graphs are generated for the Collab dataset, such that Claim 2 is only partially reproduced.

Table 1: Average boundary probabilities with standard deviation for 500 graphs generated by GNNBoundary versus with the random baseline. A near-boundary graph should satisfy $p(c_1), p(c_2) \in [0.45, 0.55]$.

| Dataset | $c_1$ | $c_2$ | GNNBoundary | | | Random Baseline | |
|---|---|---|---|---|---|---|---|
| | | | Complexity | $p(c_1)$ | $p(c_2)$ | $p(c_1)$ | $p(c_2)$ |
| Motif | House | HouseX | 0.2964 | $0.612 \pm 0.391$ | $0.375 \pm 0.389$ | $0.17 \pm 0.120$ | $0.097 \pm 0.150$ |
| | House | Comp_4 | 0.2319 | - | - | $0.00 \pm 0.002$ | $0.950 \pm 0.199$ |
| | HouseX | Comp_5 | 0.0833 | $0.528 \pm 0.262$ | $0.436 \pm 0.229$ | $0.939 \pm 0.219$ | $0.060 \pm 0.219$ |
| Collab | HE | CM | 0.0247 | $0.462 \pm 0.05$ | $0.476 \pm 0.008$ | $0.853 \pm 0.240$ | $0.143 \pm 0.241$ |
| | HE | Astro | 0.1055 | $0.513 \pm 0.023$ | $0.425 \pm 0.041$ | $0.512 \pm 0.446$ | $0.479 \pm 0.451$ |
| Enzymes | EC1 | EC4 | 1.27e-4 | $0.427 \pm 0.008$ | $0.446 \pm 0.01$ | $0.139 \pm 0.190$ | $0.281 \pm 0.346$ |
| | EC1 | EC5 | 1.51e-3 | $0.543 \pm 0.041$ | $0.361 \pm 0.046$ | $0.230 \pm 0.291$ | $0.315 \pm 0.310$ |
| | EC1 | EC6 | 2.59e-3 | $0.481 \pm 0.078$ | $0.373 \pm 0.057$ | $0.153 \pm 0.187$ | $0.180 \pm 0.216$ |
| | EC2 | EC3 | 0.0198 | $0.328 \pm 0.088$ | $0.396 \pm 0.121$ | $0.167 \pm 0.216$ | $0.225 \pm 0.252$ |
| | EC4 | EC5 | 0.0145 | $0.277 \pm 0.114$ | $0.717 \pm 0.112$ | $0.260 \pm 0.314$ | $0.387 \pm 0.321$ |
| | EC5 | EC6 | 0.1150 | $0.513 \pm 0.116$ | $0.467 \pm 0.122$ | $0.382 \pm 0.291$ | $0.239 \pm 0.242$ |
| Proteins | Non-Enzyme | Enzyme | 0.0872 | $0.551 \pm 0.040$ | $0.449 \pm 0.040$ | $0.816 \pm 0.270$ | $0.184 \pm 0.281$ |

**Boundary Analysis** The results of the boundary analysis (Figure 2) diverges from the original study (Appendix C, Figure 7), likely attributed to sampling from two models: GNNInterpreter and GNNBoundary. Both are generative models which inherently introduces randomness, in addition to requiring hyperparameter tuning to achieve convergence. For the Collab dataset, most results are at least directionally consistent with the original study, except for the margin between High Energy and Condensed Matter and the thickness between Astro and High Energy. In contrast, the Enzymes dataset exhibits much larger values for both boundary margin and thickness, similarly to how margin complexity is also very different to the original study as seen in Appendix C, Table 4.

The complexity scores show notable variability across all datasets relative to the original study. For the Motif dataset, the complexity is orders of magnitude higher. The Collab dataset shows more modest deviations, while Enzymes yields lower complexity scores than the original study. These discrepancies underscore the sensitivity of boundary analysis metrics to dataset characteristics as well as the boundary graphs produced by GNNInterpreter.

## 6.2 Results beyond original paper

GNNBoundary performs exceptionally well on the Proteins dataset, with a high training success rate of 99.3% (Appendix D, Table 5) as well as generation of faithful boundary graphs within the 0.45-0.55 range with a small variance seen in Table 1.

When performing the boundary analysis, we lack direct results from the original paper for comparison. Instead, we assess the validity of the results by comparing values relative to each other. Both the margin and thickness are larger when using the GNNInterpreter graphs from the Non-Enzyme class relative to the Enzyme class (Figure 4). Plotting the principal components of the GCN embeddings (Figure 3) reinforces that this should indeed be the case, given that the boundary graphs are more concentrated in an area of high density for Enzymes. This implies smaller margin and thickness values, which supports the values we see in Figure 4. This further supports the efficacy of GNNBoundary in explaining the decision-making of GCNs.

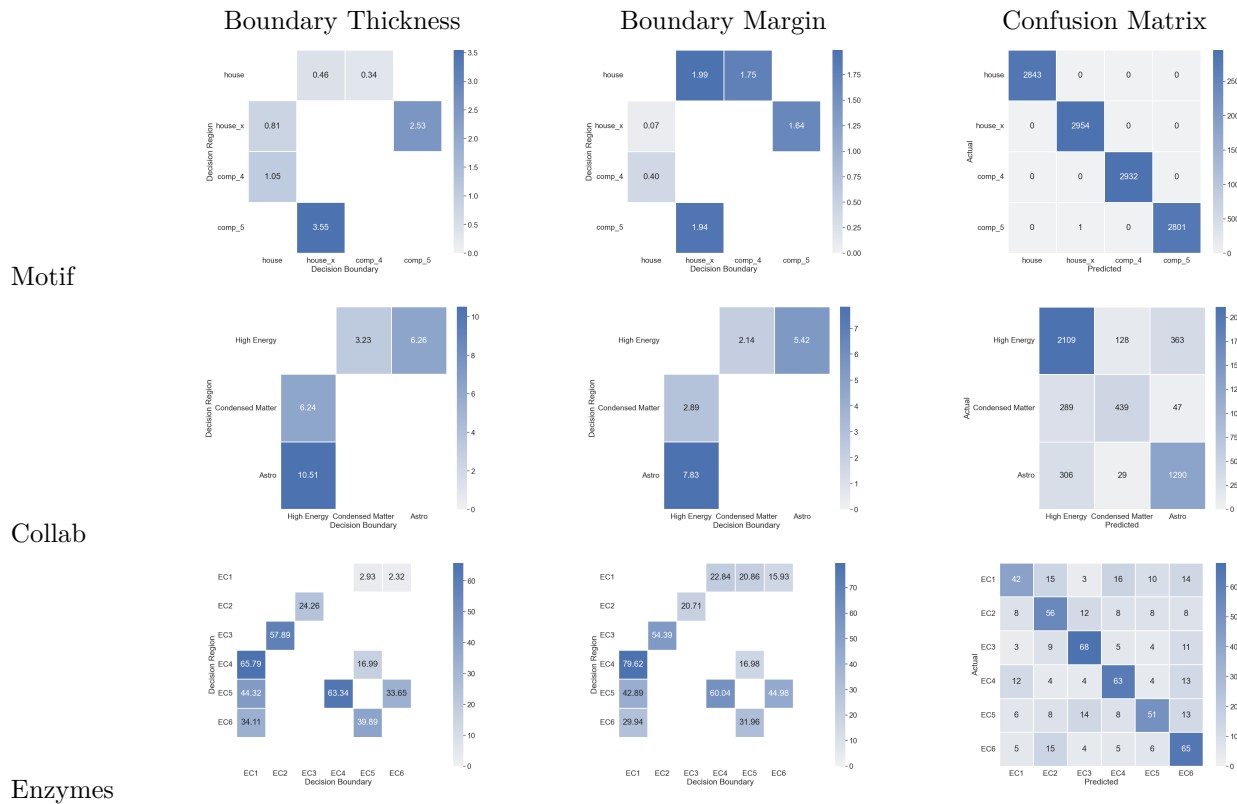

Figure 2: Boundary Thickness, Boundary Margin and Confusion Matrix of Motif, Collab and Enzymes Datasets.

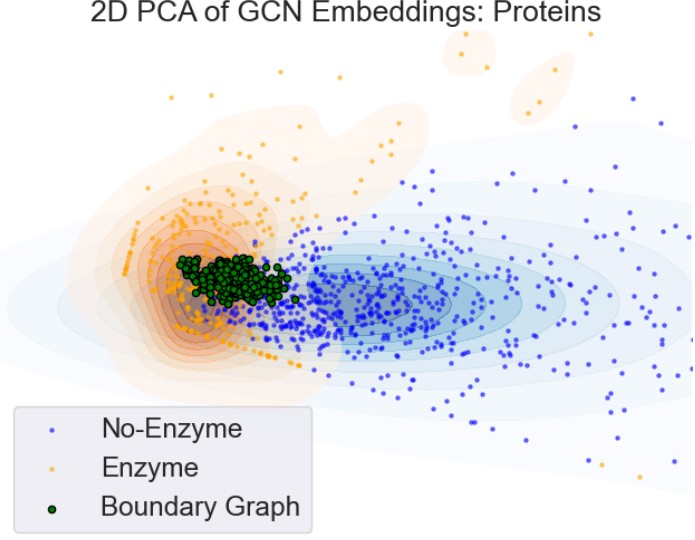

Figure 3: 2D PCA projection of GCN embeddings of the decision regions of the Proteins classifier along with 200 sampled GNNBoundary graphs. The large overlap of the boundary graphs with a high-density area of enzymes suggests a thinner boundary towards the Enzyme class.

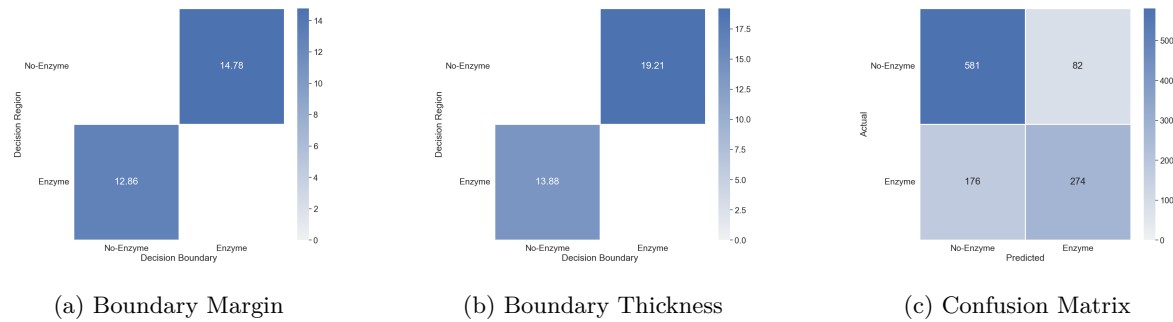

(a) Boundary Margin        (b) Boundary Thickness        (c) Confusion Matrix

Figure 4: Boundary Margin, Boundary Thickness and Confusion Matrix for the Proteins Dataset. Bottom left values in (a) and (b) were compared against GNNInterpreter samples for the Enzyme class, while the top-right scores used Non-Enzyme samples.

## 7 Discussion

In this study, we conducted several experiments to validate and investigate the main claims of 'GNNBoundary: Towards explaining Graph Neural Networks through the lens of decision boundaries' (Wang & Shen, 2024). We achieved partial replication on the three most significant claims, and highlighted variations in our experimental setup and results.

We first verified whether it is possible to find the same adjacent class pairs using the proposed method (Appendix A, Algorithm 1). Our results indicated Claim 1 is only partially supported, highlighting variabilities inherent in this method and suggesting identification may be more robust for some datasets (e.g., Collab) than others (e.g., Motif and Enzymes).

For each of the adjacent class pairs specified in the original study, we followed the GNNBoundary training procedure (Appendix A, Algorithm 2) to learn to generate faithful boundary graphs. This process required extensive hyperparameter tuning (Appendix B, Table 2) to achieve convergence on most class pairs, partly due to a lack of clear documentation of hyperparameters. More importantly, however, the proposed model was found to suffer from high variance, with low success rates for many pairs (Appendix D, Table 5) and even 0 for a Motif pair. Generating boundary graphs is ultimately also done via sampling — which can further impede robustness. During our experiments, convergence did not always guarantee only faithful boundary graphs would be sampled. We see this especially for Enzymes, which was the most complex dataset in our experiments.

Despite these challenges, we were able to on average generate boundary graphs for most pairs, finding partial support for Claim 2. Our exploration of the Proteins dataset also suggests GNNBoundary can accommodate more sophisticated real graph data when the decision landscape is simpler, i.e. when there are few decision regions.

Furthermore, we studied how GNNBoundary samples can be used to gain insights into the decision boundary of the GCN. In contradiction to the original findings, we determined Enzymes not to have higher boundary complexity values than Motif or Collab. This outcome indicates GNNBoundary may be an unreliable explainability tool for datasets with many classes. Result variations were observed across all (original) datasets, but some trends were preserved. However, we were able to deepen our analysis of these metrics with a qualitative analysis of the decision boundary between the two classes in Proteins (Figure 3). We found the 2D PCA projection of the decision regions with sampled boundary graphs aligns with the boundary margin and thickness values between GNNInterpeter samples and GNNBoundary graphs (Figure 4).

Although explainability via boundary margin, thickness, and complexity may be limited, boundary graphs can also provide a simplified, topological representation of complex structures, allowing for the identification of key structural patterns that differentiate between classes. We see that indeed, GNNBoundary samples for the Proteins dataset are structurally similar to both enzymes and non-enzymes, but abstract away from biochemical details (Figure 5). The interpretability of individual GNNBoundary samples requires further

precise and comprehensive exploration in future research, but this approach shows promise in improving model alignment with human expectations. Interestingly, boundary graphs could be moreover applied in adversarial training to improve model robustness. This application would be in line with the original proposals of these metrics by Yang et al. (2020) and Guan & Loew (2020).

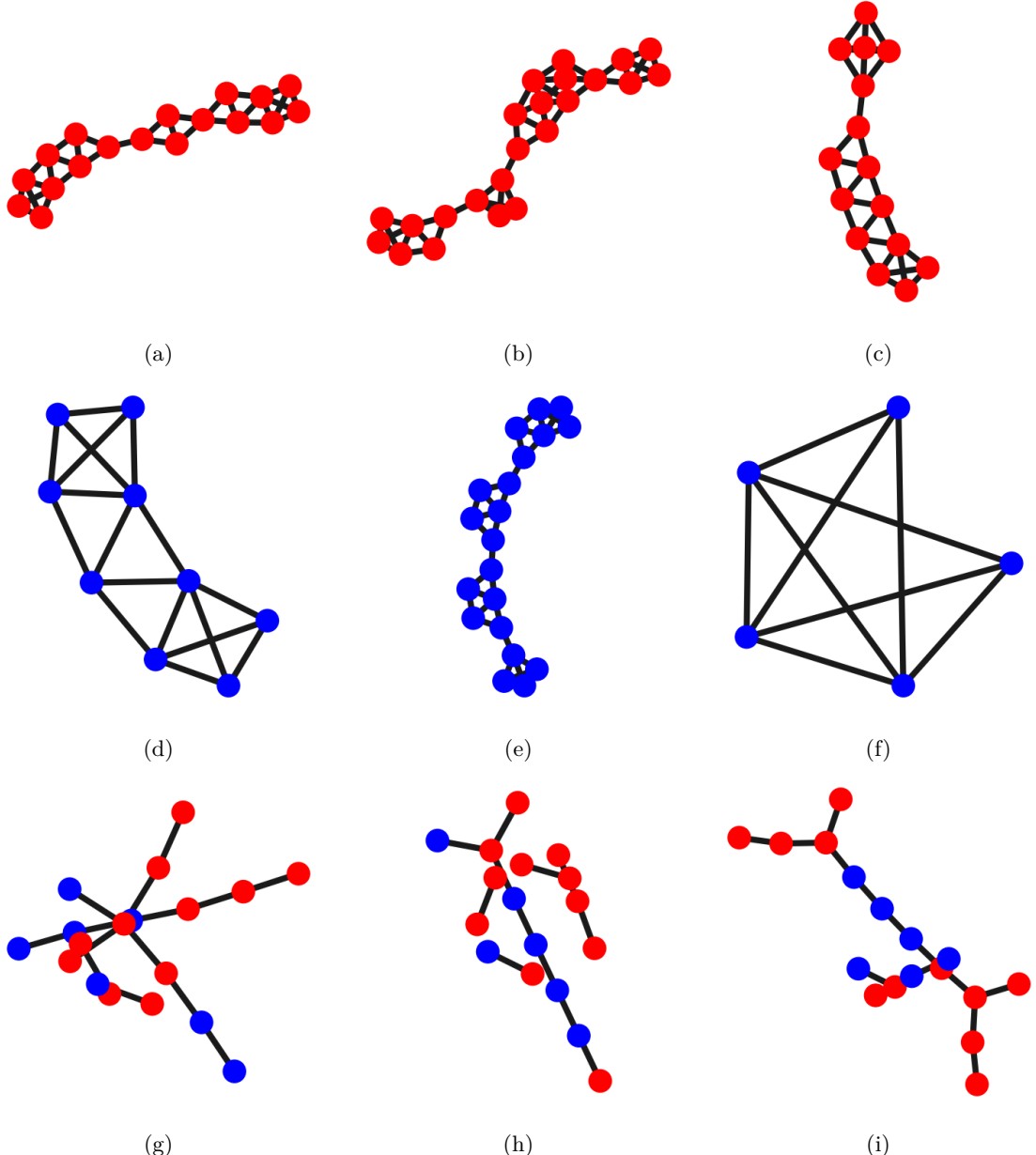

Figure 5: Comparison of Non-Enzyme (Red) and Enzyme (Blue) Proteins with GNNBoundary samples. (a–c) Non-enzyme proteins shown in red. (d–f) Enzyme protein structures shown in blue. (g–i) Generated boundary graphs combining both enzyme and non-enzyme structures. The GNNBoundary graphs in the last row capture the general connectivity and topology of the original structures but differ in fine details and local interactions, simplifying key structural or functional features.

### 7.1 Reflection: What was easy? What was difficult?

Our work was facilitated by the original codebase and the easy access to the required datasets. Additionally, Wang & Shen (2024) clearly elaborate on the theoretical basis of GNNBoundary in their paper, which further supported our efforts.

Nevertheless, we encountered several challenges and limitations. One important hurdle was the dependency on GNNInterpreter. The authors propose to draw such samples for both the pair adjacency analysis as well as for the boundary descriptors. Due to similar issues that we also encountered with GNNBoundary, and because only Motif was used in the GNNInterpreter study, hyperparameter tuning and training for GNNInterpreter were imperative. Moreover, GNNInterpreter has not been found to be sufficiently robust (Vasilcoiu et al., 2024). With this in mind, we chose not to use it where it was not necessary (for class adjacency), but still depended on it for the boundary analysis.

To conduct the experiments described in the original study, we initially used the hyperparameters presented in the paper, along with the provided checkpoints for the pretrained GCN classifier models. However, the results we obtained with these settings showed discrepancies compared to those originally reported. The hyperparameter tuning of GNNBoundary was especially time-consuming, as the same hyperparameters do not always work for all pairs of adjacent classes. Moreover, refactoring the code and adding our experimental setup was challenging because of poor code documentation.

### 7.2 Communication with original authors

We contacted the original authors via email for clarifications on their paper, code, and discrepancies between the two. Specifically, we wanted to confirm GNNInterpreter graphs were used in the class adjacency analysis and ask if it might have been used for the random baseline. Moreover, we asked about unreported hyperparameter values as well as the mismatch in values mentioned in the paper versus found in the codebase. Another important question that we included, was if they could provide us with their implementations of boundary margin, thickness, and complexity to check for reproducibility. Unfortunately, we have not received any response.

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

# 8 Appendix

## A Algorithms

---

**Algorithm 1** Measure the Degree of Adjacency of a Class Pair

---

1: $\texttt{count} \leftarrow 0$
2: **for** $k \leftarrow 1 \ldots K$ **do**
3:      Randomly sample two graphs $G_{1_k} \in \mathcal{R}_{c_1}$ and $G_{2_k} \in \mathcal{R}_{c_2}$
4:      Compute $\texttt{score} = \prod_{\lambda \in [0,1]} \mathbf{1}_{\{c_1, c_2\}} \left( \arg\max_c \eta_{L-1}(\lambda \phi_{L-1}(G_{1_k}) + (1-\lambda)\phi_{L-1}(G_{2_k})) \right)$
5:      **if** $\texttt{score} \neq 0$ **then**
6:           $\texttt{count} \leftarrow \texttt{count} + 1$
7: **return** $\frac{\texttt{count}}{K}$

---

---

**Algorithm 2** Training GNNBoundary with a Dynamic Regularization Scheduler.

---

1: Initialize sampler parameters $\Omega$ and $\mathcal{Z}$
2: **for** each iteration $t$ **do**
3:      Sample a batch of $k$ graphs $\{G_1 \ldots G_K\}$ from sampler with parameters $\Omega$ and $\mathcal{Z}$
4:      $\text{loss} \leftarrow \frac{1}{K} \sum_k \mathcal{L}(G_k) + w_{\text{budget}}^{(t)} \cdot R_{\text{budget}}(\Omega) + w_L \cdot R_{L_s}(\Omega, \mathcal{Z})$
5:      Minimize loss with respect to $\Omega$ and $\mathcal{Z}$
6:      **if** $\Psi(\mathbb{E}[G]) = 1$ and the size of $\mathbb{E}[G] < B$ **then**
7:           **return** $\Omega$ and $\mathcal{Z}$

---

# B  Tuned hyperparameter values

Table 2: Hyperparameter configurations tuned for GNNBoundary.

| Dataset | $c_1$ | $c_2$ | Hyperparameters | | | | | | | | | | | | | |
|---|---|---|---|---|---|---|---|---|---|---|---|---|---|---|---|---|
| | | | $nodes_{max}$ | $w_{budget}^{(init)}$ | $s_{inc}$ | $s_{dec}$ | $R_{L1}$ | $R_{L2}$ | $w_{objective-function}$ | $w_{embed-c_1}$ | $w_{embed-c_2}$ | budget_beta | budget | gamma | target_probs | target_size |
| Motif | House | HouseX | 30 | 1 | 1.3 | 0.8 | 5 | 18 | 30 | 0 | 0 | 4 | 10 | 1 | $c_1$: (0.45, 0.55), $c_2$: (0.45, 0.55) | 110 |
| | House | Comp_4 | - | - | - | - | - | - | - | - | - | - | - | - | - | - |
| | HouseX | Comp_5 | 30 | 1.4 | 1.3 | 0.8 | 1 | 1 | 30 | 0 | 0 | 1.8 | 18 | 1 | $c_1$: (0.45, 0.55), $c_2$: (0.45, 0.55) | 150 |
| Collab | HE | CM | 25 | 1 | 1.3 | 0.8 | 1 | 1 | 30 | 0 | 0 | 1 | 10 | 1 | $c_1$: (0.45, 0.55), $c_2$: (0.45, 0.55) | 60 |
| | HE | Astro | 25 | 1 | 1.3 | 0.8 | 1 | 1 | 30 | 0 | 0 | 1 | 10 | 1 | $c_1$: (0.45, 0.55), $c_2$: (0.45, 0.55) | 60 |
| Enzymes | EC1 | EC4 | 35 | 1.4 | 1.3 | 0.8 | 1 | 1 | 8000 | 5 | 5 | 0.8 | 24 | 1 | $c_1$: (0.4, 0.6), $c_2$: (0.4, 0.6) | 250 |
| | EC1 | EC5 | 35 | 1.4 | 1.3 | 0.8 | 3 | 1 | 5000 | 10 | 10 | 0.8 | 10 | 1 | $c_1$: (0.4, 0.6), $c_2$: (0.4, 0.6) | 200 |
| | EC1 | EC6 | 35 | 1.4 | 1.2 | 0.8 | 2 | 1 | 5000 | 20 | 20 | 1.8 | 10 | 1 | $c_1$: (0.4, 0.6), $c_2$: (0.4, 0.6) | 250 |
| | EC2 | EC3 | 35 | 1.4 | 1.3 | 0.8 | 5 | 2 | 30 | 0 | 0 | 0.6 | 24 | 1 | $c_1$: (0.4, 0.6), $c_2$: (0.4, 0.6) | 150 |
| | EC4 | EC5 | 35 | 1.4 | 1.3 | 0.8 | 5 | 2 | 30 | 0 | 0 | 0.6 | 24 | 1 | $c_1$: (0.4, 0.6), $c_2$: (0.4, 0.6) | 150 |
| | EC5 | EC6 | 35 | 1.4 | 1.3 | 0.8 | 5 | 2 | 30 | 0 | 0 | 0.6 | 24 | 1 | $c_1$: (0.45, 0.55), $c_2$: (0.45, 0.55) | 150 |
| Proteins | Non-Enzyme | Enzyme | 35 | 1 | 1.1 | 0.95 | 1 | 1 | 25 | 0 | 0 | 1 | 10 | 1 | $c_1$: (0.45, 0.55), $c_2$: (0.45, 0.55) | 30 |

Table 3: Hyperparameter configurations tuned for GNNInterpreter.

| Dataset | class | Hyperparameters | | | | | | | | | | | | | | | | | |
|---|---|---|---|---|---|---|---|---|---|---|---|---|---|---|---|---|---|---|---|
| | | $nodes_{max}$ | $w_{budget}^{(init)}$ | $s_{inc}$ | $s_{dec}$ | $R_{L1}$ | $R_{L2}$ | $w_{class-criterion-maximize}$ | $w_{class-criterion-minimize0}$ | $w_{class-criterion-minimize1}$ | $w_{class-criterion-minimize2}$ | $w_{class-criterion-minimize3}$ | $w_{class-criterion-minimize4}$ | $w_{class-criterion-minimize5}$ | $w_{embed-criterion}$ | budget_beta | budget | target_probs | target_size |
| Motif | House | 30 | 1 | 1.3 | 0.8 | 1 | 1 | 50 | - | - | - | - | - | - | 5 | 0.5 | 10 | (0.9, 1.0) | 120 |
| | HouseX | 30 | 1 | 1.3 | 0.8 | 1 | 1 | 50 | - | - | - | - | - | - | 5 | 0.5 | 10 | (0.9, 1.0) | 120 |
| | Comp_4 | 30 | 1 | 1.3 | 0.8 | 1 | 1 | 50 | - | - | - | - | - | - | 5 | 0.5 | 10 | (0.9, 1.0) | 120 |
| | Comp_5 | 30 | 1 | 1.3 | 0.8 | 1 | 1 | 100 | - | - | - | - | - | - | 5 | 0.8 | 10 | (0.9, 1.0) | 120 |
| Collab | HE | 30 | 1 | 1.3 | 0.8 | 2 | 1 | 8000 | - | 10 | 100 | - | - | - | 5 | 1 | 10 | (0.9, 1.0) | 120 |
| | CM | 30 | 1 | 1.3 | 0.8 | 2 | 1 | 1000 | 1000 | - | 100 | - | - | - | 5 | 1 | 10 | (0.9, 1.0) | 200 |
| | Astro | 30 | 1 | 1.3 | 0.8 | 2 | 1 | 8000 | 10 | 100 | - | - | - | - | 5 | 1 | 10 | (0.9, 1.0) | 120 |
| Enzymes | EC1 | 30 | 1.4 | 1.3 | 0.8 | 10 | 2 | 8000 | - | 1 | 1 | 1 | 1 | 1 | 10 | 1 | 1000 | (0.9, 1.0) | 600 |
| | EC2 | 30 | 1.4 | 1.3 | 0.8 | 5 | 2 | 8000 | 1 | - | 1 | 1 | 1 | 1 | 50 | 1.2 | 1000 | (0.9, 1.0) | 600 |
| | EC3 | 30 | 1.4 | 1.3 | 0.8 | 10 | 5 | 5000 | 10 | 10 | - | 10 | 10 | 10 | 10 | 1 | 1000 | (0.9, 1.0) | 600 |
| | EC4 | 30 | 1.4 | 1.3 | 0.8 | 5 | 2 | 8000 | 1 | 1 | 1 | - | 1 | 1 | 1 | 1 | 1000 | (0.9, 1.0) | 600 |
| | EC5 | 30 | 1.4 | 1.3 | 0.8 | 10 | 5 | 3000 | 1 | 1 | 1 | 1 | - | 1 | 1 | 1 | 1000 | (0.9, 1.0) | 600 |
| | EC6 | 30 | 1.4 | 1.3 | 0.8 | 5 | 2 | 3000 | 1 | 1 | 1 | 1 | 1 | - | 1 | 1 | 1000 | (0.9, 1.0) | 600 |
| Proteins | Non-Enzyme | 35 | 1.4 | 1.3 | 0.8 | 10 | 5 | 8000 | - | 10 | - | - | - | - | 20 | 2.8 | 400 | (0.85, 1.0) | 600 |
| | Enzyme | 35 | 1.4 | 1.3 | 0.8 | 5 | 2 | 8000 | 100 | - | - | - | - | - | 20 | 1.8 | 400 | (0.9, 1.0) | 600 |

## C  Differences in quantitative evaluation

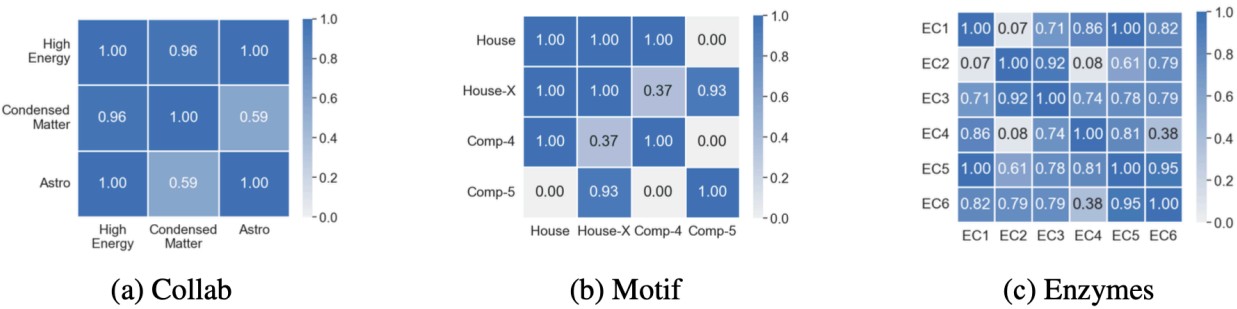

(a) Collab
(b) Motif
(c) Enzymes

Figure 6: The degree of adjacency of every class pair from Wang & Shen (2024). High values ($\geq 0.8$) indicate the decision regions of the classifier are adjacent.

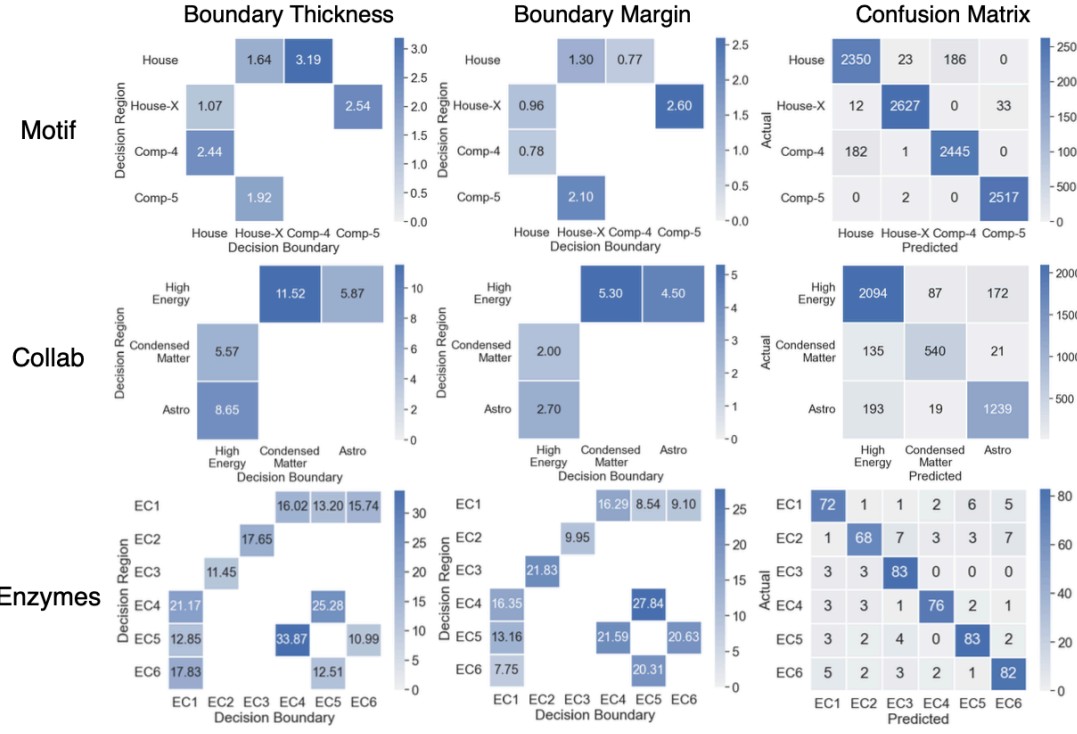

Figure 7: Boundary Margin and Boundary Thickness, as well as Confusion Matrix from Wang & Shen (2024).

Table 4: Differences in adjacent class probabilities and complexities between Table 1 of this study and Table 1 of the original study (Wang & Shen, 2024).

| Dataset | $c_1$ | $c_2$ | GNNBoundary | | |
|---------|-------|-------|-------------|--------|--------|
| | | | Complexity | $p(c_1)$ | $p(c_2)$ |
| Motif | House | HouseX | 0.2964 | 0.111 | -0.124 |
| | House | Comp_4 | 0.2319 | - | - |
| | HouseX | Comp_5 | 0.0833 | 0.037 | -0.073 |
| Collab | HE | CM | -0.0216 | -0.011 | -0.011 |
| | HE | Astro | 0.0809 | -0.013 | -0.041 |
| Enzymes | EC1 | EC4 | -0.0753 | -0.062 | -0.041 |
| | EC1 | EC5 | -0.1398 | 0.051 | -0.128 |
| | EC1 | EC6 | -0.1466 | -0.004 | -0.099 |
| | EC2 | EC3 | -0.2127 | -0.16 | -0.092 |
| | EC4 | EC5 | -0.1218 | -0.203 | 0.231 |
| | EC5 | EC6 | -0.097 | 0.032 | -0.019 |

# D  Success rate and average iteration

Table 5: Success rates and average convergence iterations for achieving near-boundary convergence within 500 iterations over 1000 runs.

| Dataset | $c_1$ | $c_2$ | Success Rate | Average Convergence Iteration |
|---|---|---|---|---|
| | | | GNNBoundary | GNNBoundary |
| Motif | House | HouseX | 0.473 | 206.920 |
| | House | Comp_4 | 0 | - |
| | HouseX | Comp_5 | 0.435 | 238.113 |
| Collab | HE | CM | 1.00 | 67.044 |
| | HE | Astro | 0.997 | 13.115 |
| Enzymes | EC1 | EC4 | 0.035 | 270.2 |
| | EC1 | EC5 | 0.166 | 90.036 |
| | EC1 | EC6 | 0.067 | 226.881 |
| | EC2 | EC3 | 0.858 | 110.969 |
| | EC4 | EC5 | 0.248 | 88.605 |
| | EC5 | EC6 | 0.975 | 24.764 |
| Proteins | Non-Enzyme | Enzyme | 0.993 | 7.021 |

