# OpenReview forum: "[Re] GNNBoundary: Towards explaining Graph Neural Networks through the lens of decision boundaries"
_TMLR — Rejected by TMLR_

### Review · Reviewer_TEpz · 2025-04-07

**Summary Of Contributions:**

This work reevaluates and reproduces a previous ICLR publication, "GNNBoundary: Towards Explaining Graph Neural Networks through the Lens of Decision Boundaries." In addition, it further evaluates the previously proposed method on an additional dataset, Proteins. The authors confirm, for the most part, the claims made in the original work.

**Audience:**

Yes

**Claims And Evidence:**

No

**Requested Changes:**

- Please use Grammarly or some other grammar checker to correct your work. Your work contains multiple sentences that are either grammatically incorrect or at least written in poor style. Examples include "For GNN with L layers, the embedding function ...", "... the outcome of the GNN for specific input.", "... input. It means that these methods address ...". (strengthens work)
- In your introduction, paragraph three, you are not providing any references for your claims. Please add a reference to each of the first two sentences, one for general model-level explanations and one for the three main limitations. (strengthens work)
- Please add a detailed discussion of related work, especially work since the publication of GNNBoundary. If you cannot fit this into the main content, feel free to add it to the appendix. (Acceptance)
- In sections 3 and later, you use f_k without defining it. (Acceptance)
- In section 3, towards the end, you use \nu instead of \nu_l. (Acceptance)
- In section 3, you use for the definition of the boundary between c_1 and _2 and greater than. I would suggest either adding a citation here or using a greater or equal than. (strengthens work)
- In section 4, the definition of Property 1 is unclear from context. It is not clear what encouraging or discouraging a probability is. Also, p(b) is not defined. Which graphs are you considering here? (Acceptance)
- In section 4, "However, to optimize the objective function, we need \Nabla_A L(G), which doesn’t exist as the graph data is discrete." It is not clear what the A is referring to here. It is likely that adjacency matrix. I would suggest explicitly stating this in some way to make this clear, like "However, to optimize the objective function, we need **the gradient with respect to the adjacency matrix** \Nabla_A L(G), which doesn’t exist as the graph data is discrete." or similar. (strengthens work)
- In section 4.4 first equation, you did not sufficiently describe the process with which G_{c_1} and G_{c_1||c_2} are determined. You likely mean graphs for which the near boundary condition from section 4.3 is true. However, this is not stated clearly. (Acceptance)
- In section 5.1, "This dataset, comprising two classes—Enzyme and Non-Enzyme—was chosen" there is a comma missing after the em dash, complicating comprehension (strengthens work)
- The biggest issue I have is with your use of the Proteins dataset and its "replacement of Enzymes" or "alternative to Enzymes." Namely, in contrast to all previously studied datasets, Proteins is a binary dataset. We would expect boundary discovery to be easier for binary datasets. Thus, your results are not surprising and not as compelling. I would suggest that you evaluate a model trained on only 2 classes of Enzymes as a comparison. If this fails, then you have a point for most of what you are saying regarding Proteins. Otherwise, it would be more reasonable to propose the use of "binarized" datasets for boundary discovery or something similar. (Acceptance)

**Strengths And Weaknesses:**

# Strengths:
- Reproduction is an important avenue of research, especially where groundbreaking work is considered, which "GNNBoundary: Towards Explaining Graph Neural Networks through the Lens of Decision Boundaries" does appear to exhibit.
- The authors evaluate the original work on an additional dataset, including a PCA evaluation of generated graphs, strengthening previously made claims.
- The paper is well structured.

# Weaknesses
- There are multiple minor grammatical mistakes
- There is essentially no related work section
- There are multiple uses of undefined variables, likely taken from the original work
- The use of the proteins dataset as an alternative to Enzymes is questionable since it is a binary dataset. Boundary discovery is expected to be far easier for binary datasets.
- The references are repeatedly using arXiv sources instead of the publishing conference. 9/21 references are arXiv sources where at least 3 are published at various ICLR conferences.

---

### Review · Reviewer_MgDP · 2025-04-07

**Summary Of Contributions:**

This paper conducts a reproducibility study of the paper titled "GNNBoundary: Towards explaining Graph Neural Networks through the lens of decision boundaries" by Xiaoqi Wang and Han-Wei Shen which was presented at ICLR 2024. The original paper is a very interesting method of doing model-level explanations for Graph Neural Networks (GNNs) with a wide range of application scenarios and evaluation settings. The main contributions of this work here is to show that the results presented in the original work are reproducible. The main insights are that the original results conducted by the authors of GNNBoundary seem to be reproducible.

**Audience:**

No

**Broader Impact Concerns:**

No concerns.

**Claims And Evidence:**

Yes

**Requested Changes:**

As pointed out in the Weakness section, the current work is too narrow for readers to get meaningful insights about the efficacy of  _GNNBoundary_. Thus, in its **current form** I don't think TMLR's Audience criterion is met.

1. **Conduct a more elaborate analysis of the hyper-parameters**. As mentioned earlier _GNNBoundary_ is a very general method, but this generality holds probably only on paper, since there are many hyper-paramters that effect the quality of the results. From a reproducibility perspective it is very much interesting to see how sensitive _GNNBoundary_ is to the hyper parameters. A list of interesting hyper parameters: **(1)** N classes in the problem (they only use 3, 4, 6), **(2)** Optimizer used for solving the optimization (they only use SGD), and **(3)** N graphs in the problem how does this number of graphs impact solving the optimization problem and/or computing the metrics.
2. **Go beyond GCNs and check the architecture robustness**: The original paper seem to run _GNNBoundary_ only with GCNs. Is this a limitation? The paper is rather general and should allow any graph neural network architecture to be analyzed. You could run experiments with alternative GNN architectures such as GATs, GINs, or potentially even GraphSAGE to see if _GNNBoundary_ is robust and finds good boundaries and boundary graphs.

**Stability of Boundary Graph Generation:** In both of the above settings, it is very important to verify that boundary graphs are correctly generated and consistently achieve the ~50%/50% predicted class probabilities. This probably is not the case since the authors in this current work are already reporting relaxations of this. One way how to do this in a visual appealing way and summarizing the results would be by plotting the three metrics proposed in the paper (thickness, margin, complexity) on the y-axis and the hyperparameter values on the x-axis. Maybe even 2-D plots are possible for showing interactions between hyper-parameters.

**Strengths And Weaknesses:**

### Strengths:
- **The underlying paper is well selected** since _GNNBoundary_ which is a very interesting method for doing graph explanations. _GNNBoundary_ could (in theory) be applied to any GNN architecture (e.g. GCNs, GINs, GATs, ...) but needs to be properly adjusted to the model and dataset. This is a very good starting point for conducting a reproducibility study.
- The paper is self-contained. It properly addresses the related background and describes the original work well.
- Minor criticism regarding typos/writing aside, the paper is generally well written.
- The authors managed to reproduce the original results (with slight variations which is to be expected) which is already an important result. The authors had to adapt the hyper parameters of _GNNBoundary_ but in the end came to similar results. This is also interesting but leads me to the biggest weakness of the work here.

### Weaknesses
- The current **scope of the reproducibility study is very limited**, making it hard to derive new insights about _GNNBoundary_. Let me elaborate. The current reproducibility study solely focuses on running the code provided by the original study and then extending the existing experiments by adding an additional dataset. This is very shallow and does add much new information compared to the original paper. The authors were able to run the original (open source) code and reproduce the results. However, they had to play around with the hyper-parameters, which is an interesting observation showing that _GNNBoundary_ is quite sensitive. In the end, the authors could reproduce the findings, but they do not discuss nor do they dig deeper into the method and how to properly interact with _GNNBoundary_. It is unclear how hyper-parameters affect _GNNBoundary_ and where the limits of the method are. Just one example, the number of classes is, of course, a very important parameter of _GNNBoundary_ as with more classes more decision boundaries exist. The original paper is showcased with a rather low number of classes (3, 4 and 6). This work now also studies the binary case (Proteins), which still stays in the regime of the original paper. What would happen when we go much higher (10, 20, or even 100 classes)? Does _GNNBoundary_ scale?
- The new dataset used here is the well known Proteins dataset, which is used throughout the GNN literature. While often studied, it also is not so interesting for doing a reproducibility study with since it is quite similar to the already used Enzymes data.
- The paper is overly verbose given the rather limited scope of the contributions.
- The visualization / summarization of the results could be improved with plots having hyper parameters on the x axis and metrics on the y-axis containing confidence bands based on independent runs with varying random seeds.

---

### Review · Reviewer_vnpX · 2025-07-23

**Summary Of Contributions:**

This paper reproduces and verifies the claims made in the paper “GNNBoundary: Towards explaining Graph Neural Networks through the lens of decision boundaries”, which examines the decision boundaries of GNN models for graph classification task. The authors use one more dataset (i.e., Proteins) beside the 3 datasets in the GNNBoundary paper. The main findings are that the claims in the GNNBoundary paper are only partially true, and the decision boundary analysis of GNNBoundary is sensitive to the datasets and parameter configurations.

**Audience:**

No

**Broader Impact Concerns:**

This is an experiment report and not suitable for publication at TMLR.

**Claims And Evidence:**

No

**Requested Changes:**

I think this an experiment report rather than research paper. It is not suitable for publication at TMLR, and thus I have no suggestions for requested changes.

**Strengths And Weaknesses:**

Strength

1.	The paper studies an important problem, i.e., understanding the decision boundaries of GNN models.

2.	The presentation is relatively clear and easy to follow.

Weakness

I am surprised that this paper is submitted as a research paper. With all due respect, I think it looks more like an experiment report. I have the following concerns regrading its value to be published as a research paper.

1.	Is validating GNNBoundary an important problem? For that to be true, GNNBoundary needs to be widely used or shown to be particularly effective for some tasks. I think this is not true, or at least the authors do not clarify this point in the paper.

2.	Beside the 3 datasets used in GNNBoundary, the authors use only 1 addition dataset, this is far from sufficient. Moreover, the authors do not explain how the new dataset differs from existing datasets (e.g., in graph style or connectivity). For an experiment validation, to be comprehensive, the datasets should cover the entire spectrum of possible properties.

3.	The observations lack insights. The conclusion seems that “the claims in the GNNBoundary paper are only partially true, and the decision boundary analysis of GNNBoundary is sensitive to the datasets and parameter configurations”. How should we revise the claims of GNNBoundary to be true? What are the reasons that cause the sensitivity of GNNBoundary to the datasets and parameter configurations? How can we avoid such sensitivity? Any guidelines on configuring the parameters?

---

### Decision · Action_Editor_yiib · 2025-09-14

**Recommendation:** Reject

**Additional Comments:**

The paper lacks in multiple aspects, including key ML insights, experimental evaluation, etc.

**Audience:**

No

**Audience Explanation:**

Given the limited ML contributions, the paper is unlikely to be of interests to the community.

**Claims And Evidence:**

No

**Claims Explanation:**

The experiments are basically applying the original code to one more dataset.